# Physics-Aware Flow Data Completion Using Neural Inpainting

## Abstract

In this paper we propose a physics-aware neural network for inpainting fluid flow data. We consider that flow field data inherently follows the solution of the Navier-Stokes equations and hence our network is designed to capture physical laws. We use a DenseBlock U-Net architecture combined with a stream function formulation to inpaint missing velocity data. Our loss functions represent the relevant physical quantities velocity, velocity Jacobian, vorticity and divergence. Obstacles are treated as known priors, and each layer of the network receives the relevant information through concatenation with the previous layer's output. Our results demonstrate the network's capability for physics-aware completion tasks, and the presented ablation studies show the effectiveness of each proposed component.

## 1 Introduction

Realistically modeling and predicting fluid phenomena is important to a large number of applications, which may range from optimizing objects' aerodynamic properties to creating special effects in movies. Fluids are commonly modelled by numerically solving the Navier-Stokes equations, however computer generated solutions might be discrepant from real phenomena. This happens possibly due to mismatches of the mathematical model, incorrect numerical discretization, poor discrete resolution, or errors on the estimation of parameters. Thus, approaches such as Particle Image Velocimetry or Doppler flow measurements directly measure fluid quantities in real-world settings. Often though, these measurements cannot be performed densely due to missing sensors, under-sampled domains or occluded and unreachable areas.

Thus, methods for prediction and augmentation of measured flow data are actively researched. Previous approaches are either based on a low-dimensional analysis of the flow field based on Principal Component Analysis (PCA), e.g., (Saini et al., 2016), or are based on physically reconstructing missing areas by solving the unsteady incompressible Navier-Stokes equations, e.g., (Sciacchitano et al., 2012). A main challenge with these traditional techniques is to predict data in large occluded or empty areas, e.g., where more than 50% of the data has to be predicted. In such challenging scenarios, already approximate predictions are useful as they could, for example, optimize strategies for sensor placement, guide procedures for human-based scanning, or improve workflows for digital prototyping.

This goal of estimating missing flow field data has many similarities with image inpainting, as it is essentially a scene completion process using partial observations. The recent success of data-driven image inpainting algorithms (Pathak et al., 2016; Iizuka et al., 2017; Liu et al., 2018; Yu et al., 2018a;b) demonstrates the capability of deep neural networks to complete large missing regions in natural images in a plausible fashion. The major difference between flow field inpainting and image inpainting lies in the fact that flow field data inherently follows the solution of the Navier-Stokes equations, and hence existing image inpainting algorithms can easily fail in physics-aware completion tasks as they never aim to capture the physical laws.

In this paper, we formulate the problem of flow data completion in large empty areas as an inpainting problem, but consider the mathematical equations that model the underlying fluid phenomena in the design of the network architecture and loss functions. By synergistically combining deep learning with fluid dynamics, we are able to inpaint data in large and irregular areas. We evaluate our proposed architectures and loss functions using thorough ablation studies both quantitatively and qualitatively. The contributions of this paper can be summarized as:

- A DenseBlocks U-Net network architecture based on a stream function formulation to in-paint velocity values;

- A set of physics-derived loss functions representing velocity, velocity gradient, divergence and vorticity;

- A simple but effective way of handling solid obstacles in the learning process.

## 2 RELATED WORK

Predicting missing data for incompressible Navier-Stokes equations has been studied in the Computational Fluid Dynamics (CFD) field. Sciacchitano et al. (2012) solves the unsteady Navier-Stokes equations locally, and dimensionality reduction approaches such as proper orthogonal decomposition are applied (Venturi & Karniadakis, 2004; Higham et al., 2016; Saini et al., 2016). Data completion of large occluded and empty areas are difficult to handle with these methods, and their running time might become prohibitively expensive depending on the application.

We adapt the idea of image inpainting, which has been intensively studied in the field of learning, to reconstruct missing flow data. (Pathak et al., 2016) used Context Encoders as one of the first attempts for filling missing image data with a deep convolutional neural network. CNN-based methods are attractive due their ability to reconstruct complex functions with only few sparse samples while being highly efficient. The follow-up work by Iizuka et al. (2017) proposes a fully convolutional network to complete rectangular missing data regions. The approach, however, still relies on Poisson image blending as a post-processing step. Yu et al. (2018b) introduces contextual attention layer to model long-range dependencies in images and a refinement network for post-processing, enabling end-to-end training. Zeng et al. (2019) extends previous work by extracting context attention maps in different layers of the encoder and skip connect attention maps to the decoder. These approaches all include adversarial losses computed from a discriminator (Goodfellow et al., 2014) in order to better reconstruct visually appealing high frequency details. However, high frequency details from adversarial losses can result in mismatches from ground truth data (Huang et al., 2017), which can potentially predict missing data that diverge from physical laws. Liu et al. (2018) designs partial convolution operations for image inpainting, so that the prediction of the missing pixels is only conditioned on the valid pixels in the original image. The operation enables high quality inpainting results without adversarial loss.

Inpainting approaches have also been successfully used for scene completion and view path planning using data from sparse input views. Song et al. (2017) uses an end-to-end network SSCNet for scene completion and Guo & Tong (2018) a view-volume CNN that extracts geometric features from 2D depth images. Zhang & Funkhouser (2018) presents an end-to-end architecture for depth inpainting, and Han et al. (2019) uses multi-view depth completion to predict point cloud representations. A 3D recurrent network has been used to integrate information from only a few input views (Choy et al., 2016), and Xu et al. (2016) uses spatial and temporal structure of sequential observations to predict a view sequence.

Neural networks have also recently been applied to fluid simulations. Applications include prediction of the entire dynamics (Wiewel et al., 2019), reconstruction of simulations from a set of input parameters (Kim et al., 2019b), interactive shape design (Umetani & Bickel, 2018), inferring hidden physics quantities (Raissi et al., 2018), and artistic control for visual effects (Kim et al., 2019a). A complete overview of machine learning for fluid dynamics can be found in Brunton et al. (2020).

## 3 LEARNING FLOW DATA

Our model is inspired by Liu et al. (2018) from image inpainting. The inpainting task in image space can benefit from the capability of deep neural networks to learn semantic priors in an end-to-end fashion. To reconstruct missing flow data, however, inherent laws of fluid dynamics should be considered by the neural network. In this section we detail the proposed physically-derived architecture and loss functions incorporated into our model, which results in improved fluid reconstruction results.

### 3.1 Physics-aware Network

Our goal is to train a network that can fill empty regions of incompressible velocity fields. The input scheme is similar to standard image inpainting tasks. Given a 2D velocity field $\mathbf{u}_{in}$ with a missing fluid data region represented by a binary mask $M$ (0 for empty and 1 for known regions), the network predicts the velocity field with the same dimensionality as the input $\mathbf{u}_{out}$. We adapt the U-Net (Ronneberger et al., 2015) by adding Dense Blocks (Huang et al., 2017) at the bottleneck and replacing the normal convolution operations with modified partial convolutions (Liu et al., 2018). We modified the original scaling factors of previous modified partial convolutions to the mean of the binary mask $\overline{M}$, which leads to sharper velocity profiles in the reconstructed field. The modified partial convolution at every location is defined as

$$x' = \begin{cases} W^T(X \odot M)\overline{M} + b & if \sum(M) > 0, \\ 0 & otherwise, \end{cases} \tag{1}$$

where $W$ are the convolution filter weights, $X$ are the feature values for the current convolution window, $b$ are biases and $\odot$ denotes element-wise multiplication.

Additionally to the velocity prediction network (velocity branch), we implement a second network that directly predicts stream function values (stream function branch). This formulation can help to enforce incompressibility of the predicted velocity field and thus guide the network to predict velocities in a physically-aware manner, since $\nabla \cdot (\nabla \times \Psi) = 0$. The curl operator is also fully differentiable. This formulation is therefore suitable to become an output layer for reconstructing incompressible flow data. Thus, the stream function branch takes feature maps from the last and second last layers of the velocity branch as well as the inpainting mask and passes them through 4 densely connected convolution layers with swish activation functions (Ramachandran et al., 2017), outputting a stream function $\Psi(x, y)$ field. The velocity field can then be reconstructed through predicted stream functions by applying the curl operator:

$$\mathbf{u} = \nabla \times \Psi. \tag{2}$$

In 2D, the stream function becomes a scalar field, and the resulting velocity components are:

$$\mathbf{u}_x = \frac{\partial \psi}{\partial y}, \mathbf{u}_y = -\frac{\partial \psi}{\partial x}. \tag{3}$$

Both velocities from velocity branch and stream function branch are concatenated together and passed through a final prediction layer along with the inpainting mask. The final prediction layer is a single convolution layer that mixes velocity predictions from both branches. A detailed illustration of our architecture is shown in Figure 1. The exact parameters of the network can be found in Appendix A.

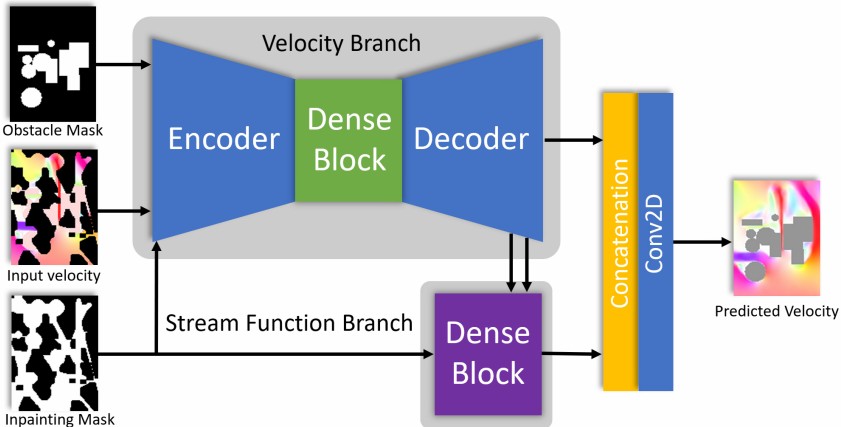

Figure 1: Our architecture for fluid data inpainting consists of a velocity branch using Dense Blocks and a stream function branch to predict incompressible velocities. Our method supports different occlusion and obstacle masks to mimic different real-world settings.

## 3.2 LOSS FUNCTIONS

For incompressible flow data it is important to define a new set of supervised loss functions that model physical properties and constraints. For detailing the employed loss functions, we use $\hat{\mathbf{u}}$ for predicted and $\mathbf{u}$ for ground-truth velocities. The $L^1$ velocity loss efficiently reconstructs low-frequency information and is defined as

$$L_{vel} = ||\hat{\mathbf{u}} - \mathbf{u}||_1. \tag{4}$$

Inspired by Kim et al. (2019b), we additionally minimize the difference of the velocity field Jacobian between ground-truth and predicted velocity fields. With a sufficiently smooth flow-field dataset, high-frequency features of the CNN are potentially on the null space of the $L^1$ distance minimization (Kim et al., 2019b). Thus, matching the Jacobians helps the network to recover high-frequency spectral information, while it also regularizes the reconstructed velocity to match ground-truth derivatives. The velocity Jacobian $J(\mathbf{u})$ is defined in 2D as

$$J(\mathbf{u}) = \begin{pmatrix} \frac{\partial \mathbf{u}_x}{\partial x} & \frac{\partial \mathbf{u}_x}{\partial y} \\ \frac{\partial \mathbf{u}_y}{\partial x} & \frac{\partial \hat{\mathbf{u}}_y}{\partial y} \end{pmatrix}, \tag{5}$$

and the corresponding loss function is simply given as the $L^1$ of vectorized Jacobian between predicted and ground-truth velocities:

$$L_{jacobian} = ||J(\hat{\mathbf{u}}) - J(\mathbf{u})||_1. \tag{6}$$

Additionally, we compute a loss function that matches the vorticity of predicted and ground-truth velocities. The vorticity field describes the local spinning motion of the velocity field. Similarly to the Jacobian loss, our vorticity loss acts as a directional high-frequency filter that helps to match shearing derivatives of the original data, enhancing the capability of the model to properly match the underlying fluid dynamics. The vorticity loss is given by

$$L_{vort} = ||\nabla \times \hat{\mathbf{u}} - \nabla \times \mathbf{u}||_1. \tag{7}$$

Incompressible flows should have zero divergence, however, numerical simulations often produce results that are not strictly divergence-free due to discretization errors. As we combine predictions from velocity and stream function branches, we are able to match the divergence on the original and predicted fields by minimizing

$$L_{div} = ||\nabla \cdot \hat{\mathbf{u}} - \nabla \cdot \mathbf{u}||_1. \tag{8}$$

Similarly to previous works, each loss function is applied both on the known and unknown regions with potentially different weights. The weight selection is illustrated in Appendix A. We exclude the perceptual $L_{perceptual}$, style $L_{style}$ and total variation $L_{tv}$ losses from the image inpainting model of Liu et al. (2018). Although these losses successfully improve the visual quality of predicted images, they are not suited for completing flow-field data, since they match pre-learned filters from image classification architectures.

## 3.3 ENCODING OBSTACLES

The interaction between fluid and solid obstacles is crucial for fluid dynamics applications. To incorporate solid obstacle information as prior knowledge to the network, we concatenate a binary mask $O$ indicating whether a solid obstacle occupies a cell as an extra input channel. In order to properly propagate the obstacle information to all network layers, the obstacle occupancy information is concatenated to previous layers' output. To accomplish that, we downsample the obstacle map $O$ to match a specific layer dimensions, similarly to the empty region mask $M$.

## 4 EXPERIMENTS AND RESULTS

### 4.1 DATA GENERATION

Due to the lack of publicly available flow data sets captured from real-world experiments, we trained our model on synthetic data. We generated fluid velocity fields with a numerical flow solver for

| | $\Psi$ branch | $\mathbf{u}$ branch | DenseBlocks | $L_{jacobian}$ | $L_{vort}$ | $L_{div}$ | MAE |
|---|---|---|---|---|---|---|---|
| (a) | ✓ | ✓ | ✓ | ✓ | ✓ | ✓ | 0.268 |
| (b) | ✓ | | ✓ | ✓ | ✓ | | 0.270 |
| (c) | | ✓ | ✓ | ✓ | ✓ | ✓ | 0.261 |
| (d) | ✓ | ✓ | | ✓ | ✓ | ✓ | 0.353 |
| (e) | ✓ | ✓ | ✓ | ✓ | ✓ | | 0.280 |
| (f) | ✓ | ✓ | ✓ | ✓ | | | 0.276 |
| (g) | ✓ | ✓ | ✓ | | | | 0.280 |
| (h) | | ✓ | | ✓ | ✓ | ✓ | 0.307 |
| (i) | | ✓ | | ✓ | ✓ | | 0.314 |
| (j) | | ✓ | | ✓ | | | 0.332 |
| (k) | | ✓ | | | | | 0.332 |

Table 1: Ablation study configurations.

incompressible fluids (Mantaflow (Thuerey & Pfaff, 2018)). Each training data sample consists of a 2-dimensional vector field containing the velocity components $u$ and $v$ and the empty regions and obstacles masks $M$ and $O$. During training, different types of empty region masks are generated on the fly with empty to filled area region ratios that vary randomly from 25 to 99 percent. We generate three different types of masks: uniform random noise masks mimic possible sampling noise from real-world velocity measurements; scan path masks simulate paths of a velocity probing; and large region masks model large occluded areas that are not reachable by probes or measurement devices. Illustrations of types of masks can be seen in the leftmost column in Figure 3.

To evaluate the proposed architecture and loss functions, we applied our model on two different datasets, both computed on grid resolutions of $128 \times 96$. The first *wind tunnel* dataset implements a scene with turbulent flow around obstacles. We define inflow velocities at bottom and top regions of the domain, while the remaining two sides (left and right) are set as free-flow (open) boundary conditions. The inflow speed is set to random values, and obstacles (6 spheres, 6 rectangles) are randomly positioned, yielding a total of 32,000 simulation frames. The second *simple plume* dataset implements a smoke rising from a source at the bottom of a fully enclosed box, which represents a common setup in graphics applications. In this dataset, no solid obstacles are included and the source position and size are the parameters that vary across different simulations. In total, 21,000 simulation frames are present in the simple plume dataset.

## 4.2 ABLATION STUDIES

To investigate the effects of various components introduced in previous sections, we performed a series of ablation studies, with results shown in Table 1. We train and evaluate the architectures of different configurations on the wind tunnel dataset, selectively deactivating the DenseBlocks, as well as the stream function and velocity branches. We also compare the effect of the proposed loss functions by progressively adding them to different architecture configurations. Besides evaluating Mean Absolute Error (MAE) over the whole dataset, we also evaluate the model capabilities when varying masking occlusion levels in Figure 2.

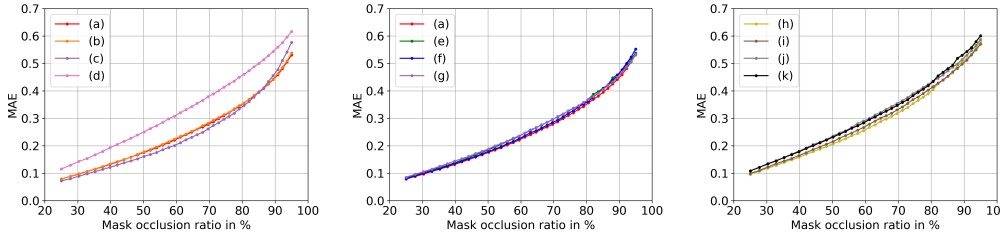

Figure 2: Mean Absolute Error of different model configurations at each mask occlusion level.

Our ablation study shows that the quality of inpainting results can be significantly improved by Dense Blocks (entries (a-c) and (e-g)) when compared with architectures with no DenseBlocks (entries (d) and (h-k)). We also tested adding loss functions progressively to architectures with (entries

(a) and (e-g)) and without DenseBlocks (entries (h-k)) to evaluate if those effects are cumulative. Combining the proposed physically-inspired losses yields better results also in this case, which seems a strong indication that they contribute to better results even in distinct architectures.

The model that employs only a stream function branch (b) performs similarly to our full model (a). Note that since model (b) reconstructs velocities by taking the curl of stream function predictions, incompressibility is guaranteed and thus no divergence loss is used. However, the synthetic velocity field data has discretization errors and it is not truly divergence free. Therefore, the approach with a single stream function branch cannot capture the divergent modes present on the original data, yielding higher MAE than the combined branches approach. We hypothesize that a pure stream function architecture would fare better with velocity fields obtained from real-world experiments or highly accurate flow solvers, where the divergence is closer to zero.

We notice that the model with only the velocity branch (c) has a lower MAE over the whole dataset. However, Figure 2 shows that this model has a low capability of inpainting samples with large empty regions. This indicates that the stream function branch can better guide the network to predict results obeying physical laws, while the velocity branch can help predicting results based on information from a known region. The functionality of the two branches can be more clearly shown in Figure 3, where the models (a,b,c) are visually compared. To plot the velocity fields, we use a HSV colormap that encodes flow direction and relative velocity in the hue and saturation, respectively. The model in (c) only uses the velocity branch information, and it reconstructs lower frequency content better as the information is easier to infer from surrounding known regions. Using the stream function branch only, model (b) better predicts high frequency information, but mistakenly reconstructs lower frequency regions. Finally, model (a) combines the advantages of models (b) and (c) by using both branches, and predictions are more precise in both higher and lower frequency ranges.

## 4.3 PREDICTION RESULTS

Figure 4 shows the results of applying different mask profiles for the simple plume dataset. The top left example shows a scan path mask (left), comparing the reconstructed (middle) and ground truth velocity profiles (right). Even with a sparse scan mask with a occlusion of $0.74$, our reconstruction is able recreate velocity profiles that are close to the ground truth, even in regions that are far away from the mask. The top and bottom right images of Figure 4 show similar examples, while the bottom left image depicts a random noise mask to mimic the effects of noisy sampling. In Figure 5 we show corresponding results for the wind tunnel dataset. In this challenging scenario, we use large region masks in combination with obstacles immersed in the fluid. Our results show that our method is able to accurately capture flows around obstacles, even though very sparse velocity field samples were used. We additionally compare the original image inpainting model (Liu et al., 2018) (Table 1 (k)) with our best architecture (Table 1 (a)). The results are shown in Figure 6, indicating that our approach can reconstruct more flow structures, especially near immersed obstacles.

## 5 CONCLUSIONS

We have presented a physics-aware architecture for inpainting missing velocity data. We have shown that our method is especially powerful for data completion of large areas with more than 50% missing data entries. Using the proposed stream function branch in combination with DenseBlocks has proven to be the key element to reduce Mean Absolute Error (MAE), and augmenting the architectures with our physically-derived loss functions has further improved accuracy. The proposed method outperforms existing image inpainting models when applied to flow data, demonstrating the effectiveness of including knowledge about fluid dynamics in the network design.

We have evaluated our method on 2-dimensional data. Extending the method to 3-dimensional flows that exhibit more turbulent structures is an essential next step. The major challenge with 3D data, however, is the large memory consumption, which is especially critical for high-resolution simulations. Approaches based on progressive patch-based inpainting (Isola et al., 2016) or view-by-view inpainting (Han et al., 2019) could be relevant for reducing the memory footprint. Further tests are also needed with simulation datasets that capture different real-world scenarios, as well as data from real-world measurements.

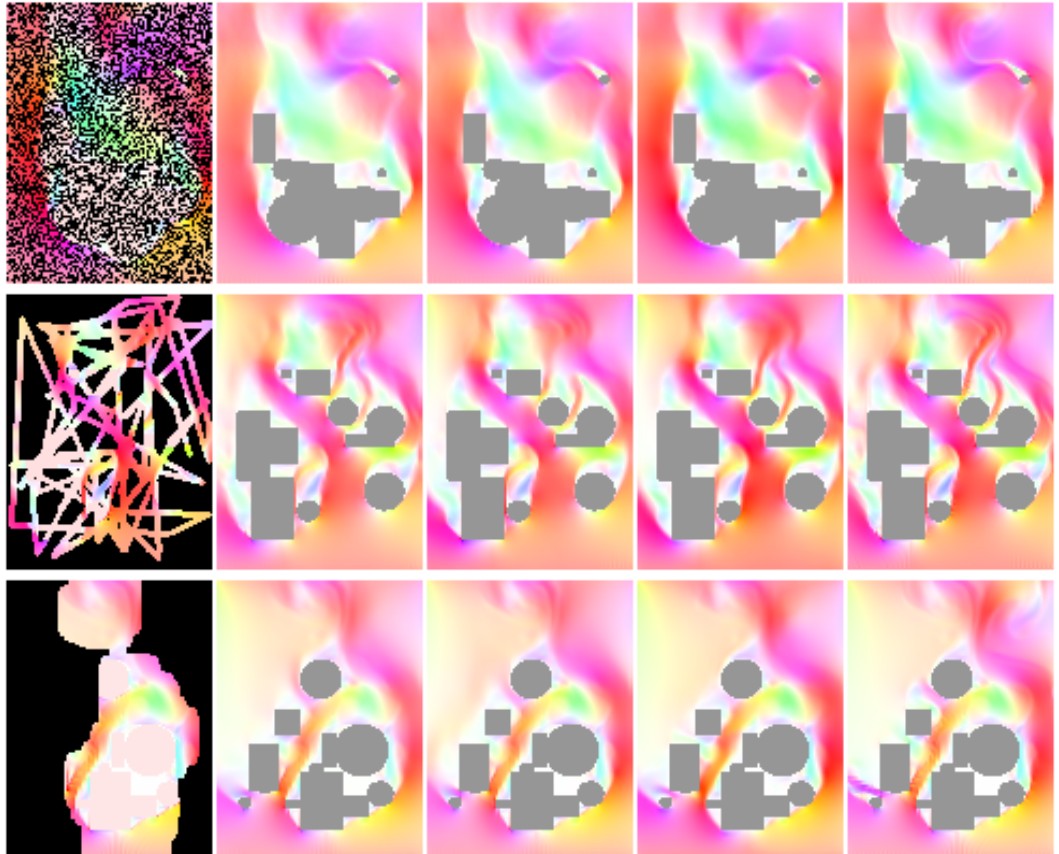

Figure 3: Inpainted flow fields with different masked inputs and obstacle configurations. From left to right: masked velocity input , output from configuration (a), output velocity from configuration (b), output from configuration (c), ground truth. All the masks used on the examples above have an approximate occlusion rate of 52%.

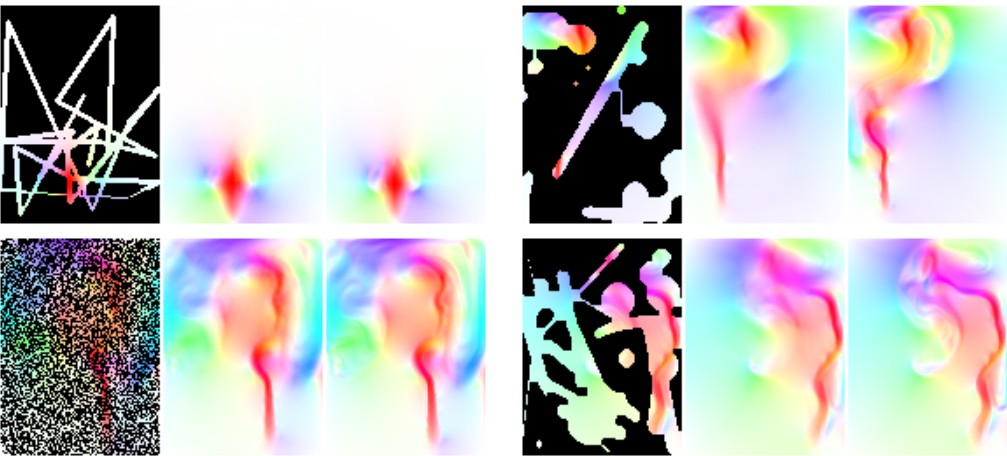

Figure 4: Examples of predictions on Simple Plume dataset. From left to right of each example: masked velocity input, output from configuration (a), ground truth velocity.

## REFERENCES

Steven L. Brunton, Bernd R. Noack, and Petros Koumoutsakos. Machine learning for fluid mechanics. *Annual Review of Fluid Mechanics*, 52, 2020.

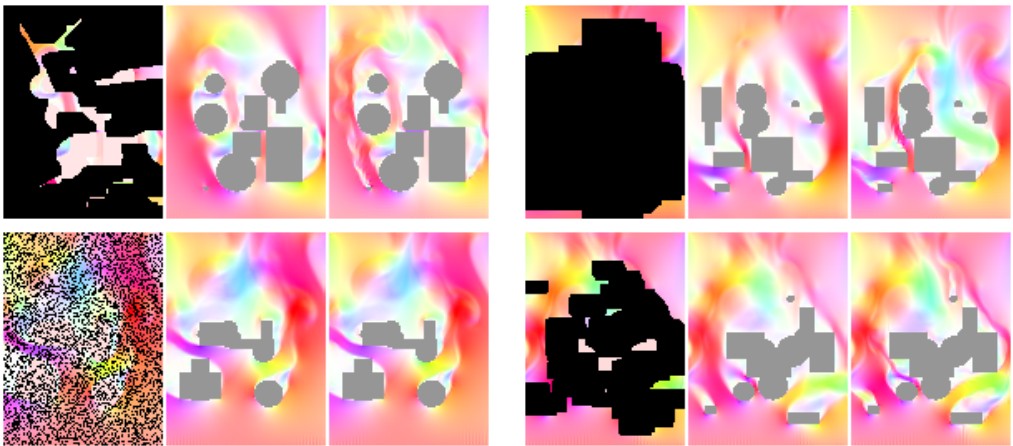

Figure 5: Examples of predictions on Wind Tunnel Dataset. From left to right of each example: masked velocity input, output from configuration (a), ground truth velocity.

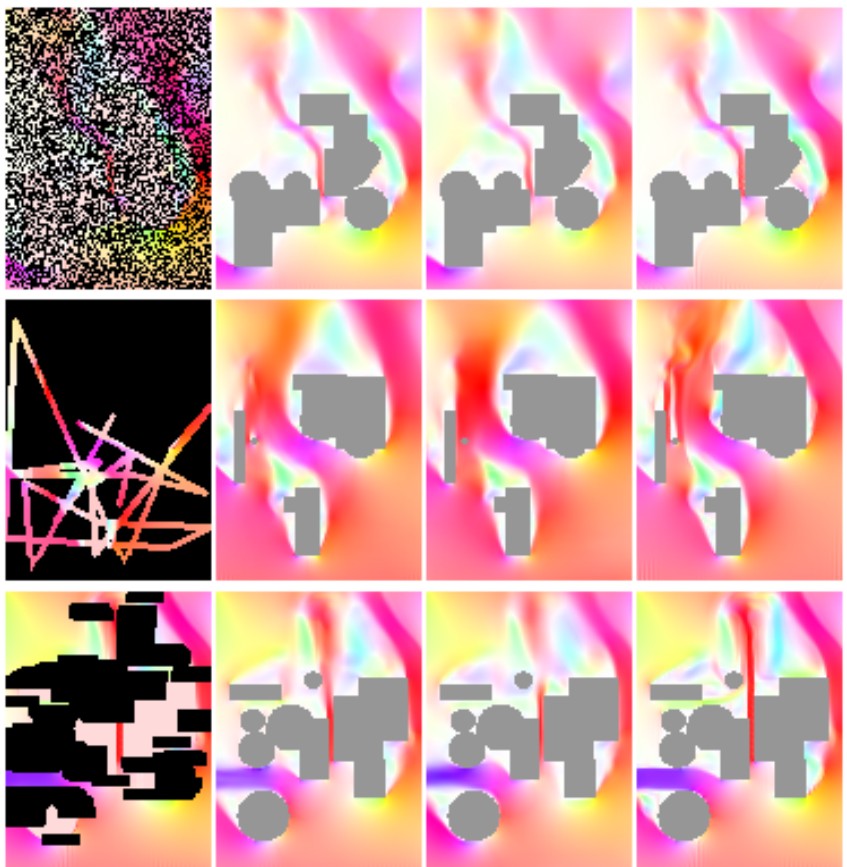

Figure 6: Comparison of model configuration (a) with image inpainting model Liu et al. (2018). Image inpainting model is trained on *wind tunnel* dataset without $L_{style}$, $L_{perceptual}$ and $L_{tv}$. From left to right: masked velocity input, output from configuration (a), output from image inpainting model, ground truth velocity.

Christopher Bongsoo Choy, Danfei Xu, JunYoung Gwak, Kevin Chen, and Silvio Savarese. 3d-r2n2: A unified approach for single and multi-view 3d object reconstruction. In *Computer Vision - ECCV 2016*, pp. 628–644, 2016. doi: 10.1007/978-3-319-46484-8\_38.

Ian Goodfellow, Jean Pouget-Abadie, Mehdi Mirza, Bing Xu, David Warde-Farley, Sherjil Ozair, Aaron Courville, and Yoshua Bengio. Generative adversarial nets. In *Advances in neural information processing systems*, pp. 2672–2680, 2014.

Yuxiao Guo and Xin Tong. View-volume network for semantic scene completion from a single depth image. pp. 726–732, 07 2018. doi: 10.24963/ijcai.2018/101.

Xiaoguang Han, Zhaoxuan Zhang, Dong Du, Mingdai Yang, Jingming Yu, Pan Pan, Xin Yang, Ligang Liu, Zixiang Xiong, and Shuguang Cui. Deep reinforcement learning of volume-guided progressive view inpainting for 3d point scene completion from a single depth image, 03 2019.

J. E. Higham, W. Brevis, and C. J. Keylock. A rapid non-iterative proper orthogonal decomposition based outlier detection and correction for PIV data. *Measurement Science and Technology*, 27 (12), 2016. ISSN 13616501. doi: 10.1088/0957-0233/27/12/125303.

Gao Huang, Zhuang Liu, Laurens Van Der Maaten, and Kilian Q. Weinberger. Densely connected convolutional networks. *Proceedings - 30th IEEE Conference on Computer Vision and Pattern Recognition, CVPR 2017*, 2017-January:2261–2269, 2017. doi: 10.1109/CVPR.2017.243.

Satoshi Iizuka, Edgar Simo-Serra, and Hiroshi Ishikawa. Globally and locally consistent image completion. *ACM Transactions on Graphics*, 36(4), 2017. ISSN 15577368. doi: 10.1145/3072959. 3073659.

Phillip Isola, Jun-Yan Zhu, Tinghui Zhou, and Alexei A. Efros. Image-to-image translation with conditional adversarial networks. *2017 IEEE Conference on Computer Vision and Pattern Recognition (CVPR)*, pp. 5967–5976, 2016.

Byungsoo Kim, Vinicius Azevedo, Markus Gross, and Barbara Solenthaler. Transport-based neural style transfer for smoke simulations. *ACM Trans. Graph.*, 2019a.

Byungsoo Kim, Vinicius C. Azevedo, Nils Thuerey, Theodore Kim, Markus Gross, and Barbara Solenthaler. Deep Fluids: A Generative Network for Parameterized Fluid Simulations. *Computer Graphics Forum*, 38:59–70, 2019b.

Guilin Liu, Fitsum A. Reda, Kevin J. Shih, Ting Chun Wang, Andrew Tao, and Bryan Catanzaro. Image Inpainting for Irregular Holes Using Partial Convolutions. *Lecture Notes in Computer Science (including subseries Lecture Notes in Artificial Intelligence and Lecture Notes in Bioinformatics)*, 11215 LNCS:89–105, 2018. ISSN 16113349. doi: 10.1007/978-3-030-01252-6_6.

Deepak Pathak, Philipp Krahenbuhl, Jeff Donahue, Trevor Darrell, and Alexei A. Efros. Context Encoders: Feature Learning by Inpainting. *Proceedings of the IEEE Computer Society Conference on Computer Vision and Pattern Recognition*, 2016-Decem:2536–2544, 2016. ISSN 10636919. doi: 10.1109/CVPR.2016.278.

Maziar Raissi, Alireza Yazdani, and George E. Karniadakis. Hidden fluid mechanics: A navier-stokes informed deep learning framework for assimilating flow visualization data. *CoRR*, 2018.

Prajit Ramachandran, Barret Zoph, and Quoc V. Le. Searching for Activation Functions. pp. 1–13, 2017. URL http://arxiv.org/abs/1710.05941.

Olaf Ronneberger, Philipp Fischer, and Thomas Brox. U-net: Convolutional networks for biomedical image segmentation. *Lecture Notes in Computer Science (including subseries Lecture Notes in Artificial Intelligence and Lecture Notes in Bioinformatics)*, 9351:234–241, 2015. ISSN 16113349. doi: 10.1007/978-3-319-24574-4_28.

Pankaj Saini, Christoph Arndt, and Adam Steinberg. Development and evaluation of gappy-pod as a data reconstruction technique for noisy piv measurements in gas turbine combustors. *Experiments in Fluids*, 57, 07 2016. doi: 10.1007/s00348-016-2208-7.

Andrea Sciacchitano, Fulvio Scarano, and Bernhard Wieneke. Multi-frame pyramid correlation for time-resolved PIV. *Experiments in Fluids*, 53(4):1087–1105, 2012. ISSN 07234864. doi: 10.1007/s00348-012-1345-x.

Shuran Song, Fisher Yu, Andy Zeng, Angel Chang, Manolis Savva, and Thomas Funkhouser. Semantic scene completion from a single depth image. pp. 190–198, 07 2017. doi: 10.1109/CVPR. 2017.28.

Nils Thuerey and Tobias Pfaff. MantaFlow, 2018. *http://mantaflow.com*.

Nobuyuki Umetani and Bernd Bickel. Learning three-dimensional flow for interactive aerodynamic design. *ACM Trans. Graph.*, 37(4):89:1–89:10, 2018. ISSN 0730-0301.

Daniele Venturi and George Em Karniadakis. Gappy data and reconstruction procedures for flow past a cylinder. *Journal of Fluid Mechanics*, 519:315–336, 2004. ISSN 00221120. doi: 10.1017/S0022112004001338.

Steffen Wiewel, Moritz Becher, and Nils Thuerey. Latent-space Physics: Towards Learning the Temporal Evolution of Fluid Flow. *Computer Graphics Forum*, 38, 2019.

Kai Xu, Yifei Shi, Lintao Zheng, Junyu Zhang, Min Liu, Hui Huang, Hao Su, Daniel Cohen-Or, and Baoquan Chen. 3d attention-driven depth acquisition for object identification. *ACM Trans. Graph.*, 35(6):238:1–238:14, 2016.

Jiahui Yu, Zhe Lin, Jimei Yang, Xiaohui Shen, Xin Lu, and Thomas S. Huang. Generative Image Inpainting with Contextual Attention. *Proceedings of the IEEE Computer Society Conference on Computer Vision and Pattern Recognition*, pp. 5505–5514, 2018a. ISSN 10636919. doi: 10.1109/CVPR.2018.00577.

Jiahui Yu, Zhe Lin, Jimei Yang, Xiaohui Shen, Xin Lu, and Thomas S. Huang. Free-Form Image Inpainting with Gated Convolution. 2018b. URL `http://arxiv.org/abs/1806.03589`.

Yanhong Zeng, Jianlong Fu, Hongyang Chao, and Baining Guo. Learning Pyramid-Context Encoder Network for High-Quality Image Inpainting. pp. 1486–1494, 2019. URL `http://arxiv.org/abs/1904.07475`.

Yinda Zhang and Thomas A. Funkhouser. Deep depth completion of a single RGB-D image. In *2018 IEEE Conference on Computer Vision and Pattern Recognition, CVPR 2018*, pp. 175–185, 2018. doi: 10.1109/CVPR.2018.00026.

## A  APPENDIX

### A.1  NETWORK PARAMETERS

The U-Net architecture is similar to the one described in (Liu et al., 2018) with the sole difference that stride 2 partial convolutions are only done in every 2nd layer in order to fit the dataset resolution of 128 by 96.

For the DenseBlocks U-Net, we replace layers 7 to 10 with a DenseBlock described in Table 3. Note that a DenseBlock as described in Huang et al. (2017) has a skip connection between each layer. Outputs of all previous layers inside the block are concatenated and represent the input of the current layer. The encoder part is also modified to achieve a constant compression ratio of 1.5 over each subsequent layer, see Table 2. The number of features is computed as follows: $\frac{64 \cdot 4^{\lfloor \frac{l}{2} \rfloor}}{1.5^{l-1}}$ with $l$ being the layer number from 1 to 7. The decoder then connects from layer 17 to 22 and is built symmetrically to the encoder. Note that this architecture, although deeper compared to the U-Net (22 vs 16), has only about 28% as many trainable parameters.

The stream function prediction network is a DenseBlock that consists of 5 convolution layers, and is described in Table 4. Activation is done with the swish function (Ramachandran et al., 2017) instead of the conventional ReLU because it turned out to facilitate training and improve accuracy.

Training for all models is done using Adam optimizer with a learning rate of 0.01 and a batch size of 16. During training, all masks are used equally and the occlusion is set uniformly at random to a value between 25% and 99%.

| Layer | 1 | 2 | 3 | 4 | 5 | 6 | 7 |
|---|---|---|---|---|---|---|---|
| Type | PConv | PConv | PConv | PConv | PConv | PConv | PConv |
| Output Resolution | 128x96 | 64x48 | 64x48 | 32x24 | 32x24 | 16x12 | 16x12 |
| Features | 64 | 170 | 113 | 303 | 202 | 539 | 359 |
| Kernel size | 7 | 5 | 5 | 3 | 3 | 3 | 3 |
| Stride | 1 | 2 | 1 | 2 | 1 | 2 | 1 |
| Activation | ReLU | ReLU | ReLU | ReLU | ReLU | ReLU | ReLU |
| Batch Normalization | No | Yes | Yes | Yes | Yes | Yes | Yes |

Table 2: Structure of the encoder part in the velocity branch.

| Layer | 8-16 |
|---|---|
| Type | PConv |
| Resolution | 16x12 |
| Features | 32 |
| Kernel size | 3 |
| Stride | 1 |
| Activation | ReLU |
| Batch Normalization | Yes |

Table 3: Structure of the DenseBlock in the velocity branch.

| Layer | 1 | 2-4 | 5 |
|---|---|---|---|
| Type | Conv | Conv | Conv |
| Padding | zero padding | zero padding | zero padding |
| Resolution | 128x96 | 128x96 | 128x96 |
| Features | 64 | 32 | 1 |
| Kernel size | 7 | 5 | 1 |
| Stride | 1 | 1 | 1 |
| Activation | swish | swish | swish |
| Batch Normalization | No | No | No |

Table 4: Structure of the DenseBlock in the stream function prediction network.

## A.2 LOSS FUNCTION WEIGHTS

The weights for all loss functions are listed in Table 5. Except for $L_{vort}$ and $L_{div}$, all losses are weighted differently on known and empty regions. We choose higher weight on empty regions because they are more important for final results.

| Loss Name | Loss Weight |
|---|---|
| $L_{vel_{valid}}$ | 1 |
| $L_{vel_{empty}}$ | 6 |
| $L_{jacobian_{valid}}$ | 1 |
| $L_{jacobian_{empty}}$ | 6 |
| $L_{vort}$ | 6 |
| $L_{div}$ | 1 |

Table 5: Loss functions weights. The losses $L_{vel}$ and $L_{jacobian}$ are weighted differently on empty regions (emtpy) and known regions (valid).

