# OpenReview forum: "Physics-Aware Flow Data Completion Using Neural Inpainting"
_ICLR.cc/2020/Conference — Reject_

### Official Review · AnonReviewer1 · 2019-10-21
**Official Blind Review #1**

**Rating:** 1

**Review:**

In this paper the authors adopt prior work in image inpainting to the problem of 2d fluid velocity field inpainting by extending the network architecture and using additional loss functions. Specifically, the U-net network is extended with a DenseBlock in the middle, and a separate branch of the network is added which predicts the stream function (a different representation of the velocity field which guarantees incompressibility). The additional losses are L1 for various derivatives of the flow field (Jacobian, divergence, vorticity). Experiments presented in the paper show that these new elements improve the flow field error compared to a baseline model originally developed for image inpainting. The suggested application for this model is filling gaps in experimental measurements that are missing or impossible to obtain, and where such a model could be computationally cheaper than an actual fluid solver.

The paper discusses prior work at the appropriate level of depth, and specifies hyperparameters that were used for training.

As is, I do not believe the paper fully meets the bar for novelty and potential impact for acceptance at ICLR (but it could perhaps be a good fit for a more specialized venue). I am specifically not convinced about the practical applicability of the results. The novelty of the approach also seems quite limited, and the findings do not seem particularly surprising or insightful (i.e. that extending the vanilla U-net architecture and adding losses that directly bias the network towards physically meaningful solutions is better than the baseline).

Since all training and testing data was generated from simulation, it is unclear how well the networks would cope with real world measurement noise. Furthermore, the authors note that the flow field from the simulation is sometimes not divergence-free. It is surprising that this could be a problem to a level where it could impact evaluations. If it indeed is, then perhaps it would make sense to either simulate the flows with a denser grid (inpainting could still be done on sparsified results). I found it surprising that the authors chose instead to use Eq. 8 and "force" the network to learn a flow field from the solver which is known to not be strictly physically valid.

It is also unclear to me that the masking schemes used in the experiments are relevant to actual measurements -- for instance, 2nd row of Fig. 3 or 1st row of Fig. 4 seem completely artificial. I appreciated the ablation study variants (but see also comments below for at least one more configuration that I believe should be discussed in addition to the existing ones). It would however be more informative (and important for potential practical applications) to include some sort of breakdown by mask type and flow field configuration/structures (steady, unsteady, wakes, jets, vortices, etc). The included images show the model does not capture some finer details of the velocity field (e.g. vortex structures in the last row of Fig. 3 and right column of Fig. 4; wakes behind small obstacles in Fig. 5-6, long, narrow, and fast jets in Fig. 6), but it is unclear what impact the proposed extensions (DenseBlock, losses, and stream function branch) have on these structures. Similarly, interpolating missing data points with a fairly dense input where no points are more than a few pixels apart seems like a much easier problem than filling large empty spaces, and it would make sense to do separate analyses for these cases.

Questions & suggestions for improvements:
* Has the impact of different weight combinations in A.2. been investigated? How was the ratio of 6:1 for empty:valid determined?
* The text says that in the stream function pathway, the features are passed "through 4 densely connected convolution layers". I was confused the first time I read that sentence, and only later I realized that this refers to a DenseNet-like pattern of connectivity. A reference to Huang et al. here or some other clarification would help.
* In the figures, please consider also showing the difference between the predictions and ground truth to make it easier to see which features of the flow field are predicted accurately, and which are not.
* Is there an L1 loss applied as well directly to the output of the stream prediction branch?
* What are the Reynolds numbers used in the simulations? How far from the original Re does the network generalize?
* When computing MAE, what are the units? What are the velocity magnitudes? Consider also reporting mean relative error.
* Impact of the various L1 deriv. losses seems negligible when the stream function is used, but is more visible when only velocity is being directly predicted. Please comment on why that might be.
* Why does (f) (Jacobian only) work better than (e) and (g)?
* It is not clear that that the effect of the additional L1 losses is truly cumulative. Please consider testing just (u, DenseBlock, L_div).
* It would be an interesting extension to include a non-ML baseline, which could be compared against the current results in terms of flow field quality and computational cost.
* Can the network predictions deviate from the known (non-masked) values in the input? If so, please consider including a breakdown of the evaluation that shows how much the error varies between the "valid" and "empty" areas.

**Experience Assessment:**

I have read many papers in this area.

**Review Assessment: Checking Correctness Of Derivations And Theory:**

I assessed the sensibility of the derivations and theory.

**Review Assessment: Checking Correctness Of Experiments:**

I assessed the sensibility of the experiments.

**Review Assessment: Thoroughness In Paper Reading:**

I read the paper at least twice and used my best judgement in assessing the paper.

---

### Official Review · AnonReviewer3 · 2019-10-23
**Official Blind Review #3**

**Rating:** 3

**Review:**

Summary:
This paper proposes a physics-aware variant of a U-Net network for completing missing flow field data. Most notably, the loss functions are motivated by fluid dynamics, which forces the network to remain more consistent with the governing laws.

Decision:
I found the paper and the idea very exciting. Injecting domain knowledge by forcing the output (or differentiable transformations thereof) to be consistent with the physics is a quite relevant and appealing idea, for which this paper constitutes a nice proof-of-concept. Framing the problem of completing missing flow data as an inpainting task is also original. However, the evaluation of the method does not study important aspects regarding its generalization. The description of the experimental protocol is also missing important information.

Further arguments:
- I found the discussion in Section 4.2 around method (b) not convincing. I do not understand why the network should be penalized if it does not reproduce the mistakes of the original simulator/solver. Since `div u` should be 0, why not simply penalizing ||div u|| instead of having the loss of Eqn. 8? Isn't approach (b) the approach which is most 'physics-aware' and correct from a physics point of view?
- The experiments do not highlight whether the network actually just learn the training distribution or generalize by "understanding" the physics of the problem. A compelling experiment would have been to evaluate whether a network trained on a prior family of obstacles transfer properly to a different family (e.g., training on 6 spherical and 6 rectangle obstacles, but testing on fewer/more obstacles with other shapes).
- Similarly, could the network generalize to larger/smaller inputs? Once trained, can it work on grids smaller/larger than 128x96? If not, what do you recommend to do in practice?
- The description of the experimental protocol does not specify whether the method was evaluated on independent test data. More worrisome, section 4.2 even states that the MAE for Figure 2 is computed "over the whole dataset".
- The method is not compared against any domain specific baseline.
- While quite exciting, I am not confident the contribution is original enough from an ML point of view for ICLR, although it is certainly novel for fluid dynamics.

Additional feedback:
- Some results reported in Table 1 are quite close of each other. It would have made the experiments much stronger if uncertainties were also reported and discussed.
- Fig 3: I would have liked seeing the error maps of the methods. This would have been quite helpful to better tell them apart.
- Given my comments above, I am confident an 8th page could be put to good use.


**Experience Assessment:**

I have read many papers in this area.

**Review Assessment: Checking Correctness Of Derivations And Theory:**

N/A

**Review Assessment: Checking Correctness Of Experiments:**

I carefully checked the experiments.

**Review Assessment: Thoroughness In Paper Reading:**

I read the paper at least twice and used my best judgement in assessing the paper.

---

### Official Review · AnonReviewer2 · 2019-10-29
**Official Blind Review #2**

**Rating:** 3

**Review:**

I am not an expert in recent Navier-Stokes approaches, but note that there is a lot of recent work in physics aware modeling.  Specifically the sections on e.g. loss seem to have a lot of prior work. It’s difficult for me to judge the exact amount of novelty in this paper with respect to the physics. With respects to the DL part it looks like it’s mainly minor modifications to the known U-net architecture.
* With respect to the introduction of the stream function branch, in table 1 one can observe that the error is actually higher than simply without it. The authors argue that “the synthetic velocity field data has discretisation errors and it is not truly divergence free. Therefore, the approach with a single stream function branch cannot capture the divergent modes present on the original data”. I’m not sure exactly what to think of this .. in my opinion this means they should use a more accurate flow solver for their simulation as otherwise it is hard to draw any definite conclusion here and only speculation remains.
* I guess the inpainting works decently well, as is expected from previous image inpainting literature and the problem is essentially treated as image completion. However, I can’t really visually tell much of a difference between the images shown for the different approaches/components in figures 3-6. Again, this ties back to my previous component about lack of clarity of the improvement of the introduced individual terms.
* I find it a little bit weird that there is exactly one reference prior to 2012. Fluid dynamics isn’t exactly a field that was introduced 5 years ago. Also their paper ends before the 8 page limit. I think the authors could have used the remaining space more efficiently.

I generally like the idea of including physical consistency when training to train a neural network for a respective task where this matters. I’m just not sure I have a clear take-away from this paper as the results don’t seem to carry a clear message of the proposed approaches resulting in definite improvements over more naive approaches or including only partial physical consistency.


**Experience Assessment:**

I do not know much about this area.

**Review Assessment: Checking Correctness Of Derivations And Theory:**

I assessed the sensibility of the derivations and theory.

**Review Assessment: Checking Correctness Of Experiments:**

I assessed the sensibility of the experiments.

**Review Assessment: Thoroughness In Paper Reading:**

I read the paper at least twice and used my best judgement in assessing the paper.

---

### Decision · Program_Chairs · 2019-12-19

**Decision:**

Reject

**Comment:**

The authors present a physics-aware models for inpainting fluid data. In particular, the authors extend the vanilla U-net architecture and add losses that explicitly  bias the network towards physically meaningful solutions.

While the reviewers found the work to be interesting, they raised a few questions/objections which are summarised below:

1) Novelty: The reviewers largely found the idea to be novel. I agree that this is indeed novel and a step in the right direction.
2) Experiments: The main objection was to the experimental methodology. In particular, since most of the experiments were on simulated data the reviewers expected simulations where the test conditions were a bit more different than the training conditions. It is not very clear whether the training and test conditions were different and it would have been useful if the authors had clarified this in the rebuttal. The reviewers have also suggested a more thorough ablation study.
3) Organisation: The authors could have used the space more effectively by providing additional details and ablation studies.

Unfortunately, the authors did not engage with the reviewers and respond to their queries. I understand that this could have been because of the poor ratings which would have made the authors believe that a discussion wouldn't help. The reviewers have asked very relevant Qs and made some interesting suggestions about the experimental setup. I strongly recommend the authors to consider these during subsequent submissions.

Based on the reviewer comments and lack of response from the authors, I recommend that the paper cannot be accepted.